# ADAMTS13 in *Bothrops lanceolatus* snakebite envenoming: Crude venom-induced reduction of *in vitro* enzymatic activity and clinical correlation in snakebite patients

Jonathan Florentin[1,2☯], Caroline Rapon[1☯], Olivier Pierre-Louis[1], Fatima Radouani[1,3], Prisca Jalta[1,3], Pascal Fuseau[4], Hatem Kallel[5], Remi Neviere[1]*, Dabor Resiere[1,2]

1 Cardiovascular Research Team (UR5_3 PC2E), University of the French West Indies (Université des Antilles), Fort de France, France, 2 Department of Toxicology and Critical Care Medicine, University Hospital of Martinique (CHU Martinique), Fort-de-France, France, 3 Department of Research, University Hospital of Martinique (CHU Martinique), Fort-de-France, France, 4 Department of Biology – Hematology, University Hospital of Martinique (CHU Martinique), Fort-de-France, France, 5 Tntensive Care Unit, Cayenne General Hospital, and Tropical Biome and Immunopathology CNRS UMR-9017, Inserm U, Université de Guyane, Cayenne, French Guiana

☯ These authors are contributed equally on this work.
* remi.neviere@chu-martinique.fr

## Abstract

### Background

Envenoming by *Bothrops lanceolatus*, a viperid endemic to Martinique an island in the Lesser Antilles, induces a unique clinical manifestation, i.e., thrombosis. Pathophysiological signaling leading to thrombotic events remains poorly understood. Among others, proposed mechanisms include increased expression of multimerized forms of von Willebrand factor (VWF). Prothrombotic effects of VWF are regulated by ADAMTS13, a metalloprotease which cleaves VWF multimers. Whether ADAMTS13 activity is reduced in *B. lanceolatus* envenoming has not been previously investigated.

### Methodology/Principal Findings

ADAMTS13 activity was evaluated via chromogenic assay in experimental and clinical studies. Human plasma with known ADAMTS13 activities (0.70 IU/mL) was incubated with increasing doses of *B. lanceolatus* venom. Incubation with *B. lanceolatus* venom (concentrations 10–1000 ng/mL) induced a dose-dependent reduction of ADAMTS13 activity. In our series of 46 patients bitten by *B. lanceolatus* snake, ADAMTS13 activity was determined at admission before antivenom therapy. In these patients, the median plasmatic ADAMTS13 activity was 94% (IQR: 78–122%). Ten patients (22%) displayed ADAMTS13 activity less than 75%. Compared with patients with normal ADAMTS13 activity (n = 36), those with moderately low ADAMTS13

**Data availability statement:** Data are included in Supporting Information files of the submitted manuscript.

**Funding:** This work was supported by APIDOM GIRCI SOHO - Direction Générale de l'Offre de Soins (WBOTHROPS 2019 to RN) - URL: https://www.girci-soho.fr/ and ANR Agence Nationale de la Recherche (AAPG 2025 - PRC – KariBothrops to DR) - URL: https://anr.fr/ The funders had no role in study design, data collection and analysis, decision to publish, or preparation of the manuscript.

**Competing interests:** The authors have declared that no competing interests exist.

activity (n = 10) were older, had a lower platelet count, and displayed increased concentrations of creatine kinase, fibrinogen and C-reactive protein. Multivariate linear regression retained only grade severity of envenoming as independent predictor of increase length of hospital stay, while reduced ADAMTS13 activity and increased C-reactive protein levels bordered on statistical significance.

## Conclusions/Significance

For the first time, our study provided evidence suggesting that ADAMTS13 activity is reduced in experimental *B. lanceolatus* venom exposure and patients with *B. lanceolatus* envenoming. Thanks to its role on the VWF regulation, it is suggested that reduced ADAMTS13 activity can increase the risk of thrombosis in *B. lanceolatus* envenoming by favoring the circulation of prothrombotic VWF multimers. Low ADAMTS13 activity was associated with increased length of stay in envenomed patients.

---

### Author summary

The clinical manifestations of envenoming by *Bothrops* sp. typically include local tissue injury and disturbances of hemostasis characterized by consumption coagulopathy leading to incoagulable blood. In contrast, *B. lanceolatus* envenoming can elicit thrombotic events, such as ischemic stroke and myocardial infarction. Proposed mechanisms include the combination of procoagulant and systemic inflammatory processes leading to thrombotic state. The factor von Willebrand (VWF) is an active biomolecule critical for coagulation and thrombosis processes. The amount of multimerized forms of VWF increases the risk of thrombosis. VWF is regulated by ADAMTS13, an enzyme that cleaves VWF multimers and reduced their prothrombotic effects. Whether ADAMTS13 activity is reduced in *B. lanceolatus* envenoming has not been previously investigated, while only few case reports have previously evaluated the levels of ADAMTS13 activity in human envenoming. For the first time in this study, we study provided evidence suggesting that ADAMTS13 activity is reduced in experimental *B. lanceolatus* venom exposure and in human *B. lanceolatus* envenoming. Our study results further suggest that, together with high grade severity score of envenoming and systemic inflammation, moderately reduced levels of ADAMTS13 activity are associated with increased length of hospital stays.

## Introduction

The clinical manifestations of envenoming by *Bothrops* sp. include local effects, such as pain, edema, and necrosis, and systemic complications, such as disturbances of hemostasis characterized by consumption coagulopathy and defibrinogenation, and, in some cases, acute kidney failure and circulatory shock [1–3]. In contrast to the

typical hemostatic alterations induced by *Bothrops* sp. venoms, envenoming by *B. lanceolatus* can elicit diffuse thrombotic microangiopathy and severe thrombotic events within 48h of the bite [4–6].

*B. lanceolatus* is a viperid species endemic to Martinique, an island in the Lesser Antilles of the West Indies in the south of the Caribbean Sea. Before the era of immunotherapy, *B. lanceolatus* envenoming was associated with systemic arterial thrombotic complications leading to cerebral, myocardial or pulmonary infarctions in 25–30% of cases and was fatal in about 10% of cases [7–11]. Since the mid 1990s, the use of the monospecific antivenom Bothrofav (Micropharm, UK), has reduced mortality to zero and prevented most of the thrombosis complications [7–11]. Despite excellent efficiency of Bothrofav, the exact pathophysiological mechanisms leading to *B. lanceolatus* venom-induced thrombotic complications remain poorly understood. In absence of overt proaggregating or procoagulant activities, proposed mechanisms of *B. lanceolatus* venom-induced thrombosis include increased expression of tissue factor, complement system activation, endothelial dysfunction and immune inflammatory systemic response [10,12–14].

Biological features consistent with thrombotic microangiopathy has been described after snakebite for many years [15,16]. In viperid envenoming, thrombotic microangiopathy has been previously described in some *Bothrops* sp. envenoming, including by *B. jararaca*, *B. venezuelensis* and *B. erythromelas* [17–19]. In these cases, patients presented on admission a typical local bothropic syndrome with no signs of systemic bleeding, whereas thrombotic microangiopathy can developed within the few days after snakebite [15–19]. Of note, thrombotic microangiopathy is the overarching term used to describe any disease process characterized by thrombocytopenia and microangiopathic hemolytic anemia with or without other clinical or laboratory features (thrombocytopenia, hemolytic anemia with presence of schistocytes in blood smears, decreased haptoglobin level, increased lactate dehydrogenase level and acute kidney injury) [20,21]. Primary thrombotic microangiopathies include hemolytic uremic syndrome (HUS) and thrombotic thrombocytopenic purpura (TTP). Typical TTP is broadly defined as a thrombotic microangiopathy occurring in the context of severe ADAMTS13 deficiency (<10%), while ADAMTS13 is usually not decreased in HUS [20–22]. Secondary forms of thrombotic microangiopathies, which occur in the context of pregnancy, autoimmune disease, malignancy, bone marrow transplantation or use of certain medications, are associated with mildly reduced or even normal ADAMTS13 activity [21–23].

ADAMTS13 is a disintegrin-like and metalloproteinase with a thrombospondin type 1 motif, member 13, which regulates hemostasis and thrombosis processes related to von Willebrand factor (VWF) pathway. VWF activity is regulated by ADAMTS13, an enzyme that cleaves VWF multimers and reduced their prothrombotic effects [23]. Increasing evidence suggest that even moderate decreases of ADAMTS13 activity can also predispose to ischemic cardiovascular diseases, particularly myocardial infarction and ischemic stroke [24,25]. Previous experimental reports in *Bothrops* sp. envenoming have described mild reduction of ADAMTS13 activity [26,27]. In *B. jararaca* envenoming, reduction of ADAMTS13 was associated with reduced VWF levels and cleavage of high molecular weight VWF multimers [26]. *B. jararaca* venom was also shown to downregulate mRNA synthesis of hepatic *Adamts13* [28]. In human envenoming, only few case reports have previously evaluated the levels of ADAMTS13 activity, which were normal when tested [17,18,29–31].

Whether reduced ADAMTS13 activity would play a role in the endothelial dysfunction and risk of thrombosis associated with *B. lanceolatus* envenoming have not been studied. We hypothesized that moderately reduced levels of ADAMTS13 activity may be involved in the pathophysiological mechanisms of the thrombotic complications associated with *B. lanceolatus* envenoming. Primary objectives of our study were to test whether plasmatic ADAMTS13 activity would be reduced by *in vitro* exposure to *B. lanceolatus* venom and in humans bitten by *B. lanceolatus* snake. Secondary objective of the study was to test the clinical relevancy of low ADAMTS13 levels in patients bitten by *B. lanceolatus* snake.

## Patients and methods

### Ethics statement

The study was approved by the CHU Martinique hospital's institutional review board (IRB #01072019). Patients were managed in accordance with the amended Declaration of Helsinki (http://www.wma.net/ en/30publications/10policies/b3/).

Patients were informed that their clinical and laboratory results may be used for research purposes. This consent was recorded in the patient's electronic medical record. Due to the retrospective nature of this study and anonymous analysis of patient records, the exemption of written informed consent was approved by the IRB of CHU Martinique.

### Venom and antivenom

Crude venom was obtained from 6 adult females and 6 males (112–156 cm long) caught in the wild in various locations of Martinique. Median (25th-75th percentiles) length of snakes was 168 (150–172) cm in female specimens and 158 (123–166) cm in male specimens. Venom samples were pooled, lyophilized (Freezone, Labconco, Kansas City, MO, USA) and stored at -80°C until use (stock solution 10 mg/mL). In *in vitro* experiments, *B. lanceolatus* venom was used at final concentrations ranging from 10 to 1000 ng/mL.

Bothrofav (Micropharm, UK), a preparation containing F(ab')2 fragments that have the property of neutralizing *B. lanceolatus* venom, was administered in clinical *B. lanceolatus* envenoming. These equine F(ab')2 fragments ligate venom antigens present in circulating blood to form inactive F(ab')2-antigen complexes, in turn reducing the amount of free venom in circulation. Bothrofav was used (batch J8216; protein concentration of 20.7±0.05 g/dL) at the final concentrations of 0.21 mg/mL.

### Assay of ADAMTS13 activity in *in vitro* plasmatic preparation

Chromogenic assay of ADAMTS13 activity was performed in human plasma prepared from lyophilized standard control plasma as advised by the supplier (control N 5020050, Technozym, Cryopep, France) and a commercial ELISA kit for ADAMTS13 activity (Reference 4–5450701, Technozym, Cryopep, France). The plate was incubated with GST-VWF for 1 hour and then washed three times, in accordance with the manufacturer's instructions. The samples were then incubated in the wells for 30 minutes, followed by three other washes. The plate was then incubated with the HRP-conjugated anti-VWF antibody for 1 hour and washed three times. Finally, the plate was incubated with the chromogenic substrate for 30 min and an acetic acid solution was added rapidly to stop the reaction. Absorbance was measured within 10 minutes. The optical density reading was obtained using a spectrophotometer at 450 nm (AMR-100, Allsheng, Hangzhou, China).

### Assay of ADAMTS13 activity in envenomed patients

Venous blood was collected into 5-mL blue-top tubes (3.2% sodium citrate) and kept at 4° C for less than 24 hours. Anticoagulated blood was centrifuged at 1100g for 10 minutes, and plasma was stored at -70° C until assay. Levels of ADAMTS13 activity were measured via a chromogenic ADAMTS13 activity ELISA kit (Reference 4–5450701, Technozym, Cryopep, France). The principle of the assay is related to the incubation of plasma with a fragment of VWF (GST-vWF73 substrate). The specific cleavage of this fragment by ADAMTS13 is then detected using a labeled antibody targeting the cleaved VWF fragment. The intensity of the staining is directly proportional to the amount of cleaved substrate and, therefore, to the ADAMTS13 activity in the plasma sample. According to manufacturer's instructions, incubation times for the GST-vWF73 substrate and study sample were 60 min and 30 min, respectively. Incubation times for the conjugated antibody and chromogenic substrate were 60 min and 30 min, respectively. The reaction was stopped and optical density was measured at 450 nm. Calibrators from 0 to 1 IU/mL (Cal Set 4–5450761, Technozym) and controls (Control Set 4–5450763, Technozym) of 0.70 IU/mL concentration were used for the determination of ADAMTS13 activity. Controls and calibrators have been calibrated against the 1st WHO international standard (NIBSC code 12/252) in IU/mL. Activity of ADAMTS13 was expressed as percent of controls.

### Patients

Patients with the diagnosis of *B. lanceolatus* envenoming presenting at the University Hospital of Martinique from January 2019 to January 2023 were retrospectively included in the study. Information regarding the possible use of the clinical records for research was provided to patients at time of hospitalization and consent was written in the medical file of the

patient. All patients were managed in accordance with the amended Declaration of Helsinki (http://www.wma.net/en/30publications/10policies/b3/). The study was approved by the CHU Martinique hospital's institutional review board (IRB #01072019).

## Diagnosis and antivenom therapy of *B. lanceolatus* snakebite envenoming

When seen, the culprit snake was identified based on the patient's description, photographs, or the physical examination of the captured snake. Snakebite was confirmed by the observation of two puncture wounds resulting from snake fang marks. Patient health data were collected using the Emergency DX Care software (Medasys, Dedalus, France) on hospital presentation. Data recording and patient management was performed with the help of dedicated senior toxicologist physicians of the snakebite team at the CHU Martinique. Clinical data, including age and gender of patients, date and time of the bite, anatomical site of the bite, grade of severity of envenoming, clinical manifestations and routine biological tests at admission, date and time of hospital admission, and antivenom administration were collected from the DX Care medical record of the patients. For ADAMTS13 activity assay, peripheral whole blood was collected with sodium citrate at admission All samples were stored at −80 °C, thawed at 37 °C immediately before testing, and were not refrozen. The severity of envenoming of each patient was measured according to a previously defined clinical severity scale. Development of envenoming, occurrence of adverse events, duration of hospital stays, and outcome were monitored for each patient. We excluded patients hospitalized for snakebite envenoming lasting more than 48 hours after the bite and those who did not present clinical or biological signs of envenoming. Envenomed patients were treated by intravenous infusion of antivenom Bothrofav at dose adjusted to the severity grade of the case.

## Statistical analysis

Quantitative data were presented as mean ± standard deviation (SD) or median IQR. The Shapiro-Wilk test was used to test for normal distribution of quantitative data. Median, interquartile range, and minimum–maximum ranges are described for non-normally distributed variables. Categorical data are presented as absolute values with percentages. The following tests were used for group comparisons: Student's t-test, analysis of variance ANOVA, chi-square test, and Fisher's exact test. When a significant difference was found, we identified specific differences between groups with a sequentially rejective Bonferroni procedure. Level of statistical significance was set at $p < 0.05$. In this exploratory work, univariate and multivariate linear regression models were implemented to assess the independent effect of potential predictors of length of hospital stay in the study population. All statistical analyses were conducted using IBM SPSS Statistics version 27 for Windows software. Figures were processed using Prism 6 for Windows software (Graphpad, Boston MA, USA).

# Results

### *In vitro effects of B. lanceolatus* venom on ADAMTS13 activity in plasmatic samples prepared from commercialized control human plasma

Venom incubation with plasma samples displaying a fixed ADAMTS13 activity induced optical density changes of the plasma staining, which is directly proportional to the amount of cleaved chromogenic substrate and, therefore, the ADAMTS13 activity in the sample. Incubation of plasma displaying known ADAMTS13 activities (0.70 IU/mL) with *B. lanceolatus* venom induced a dose dependent (1–1000 ng/mL) reduction of ADAMTS13 activity (Fig 1).

## Main characteristics and ADAMTS13 activity in human *B. lanceolatus* envenoming

From January 2019 to January 2023, a total of 116 patients suffering envenoming by *B. lanceolatus* were included in the study. Patient health data was carefully collected using the Emergency DX Care software (Medasys, Dedalus, France) on hospital presentation. Patient's registry, critical information such as clinical and biological presentation, envenoming grade

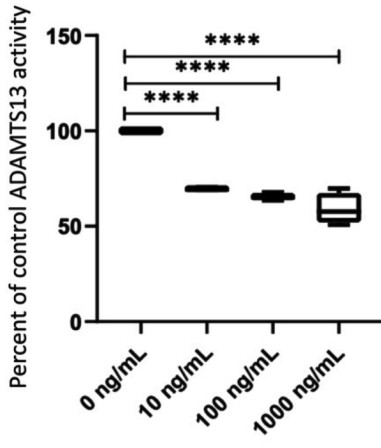

**B. lanceolatus venom**

**Fig 1. Representative ADAMTS13 activity changes following *B. lanceolatus* venom incubation with plasma samples displaying a fixed ADAMTS13 activity (0.70 IU/mL) in human plasma.** Experiments were performed with *B. lanceolatus* venom concentrations ranging from 10 to 1000 ng/mL. Results are displayed as box plot with median, min to max data points. N=5-6 experiments, *** indicates p<0.01, **** indicates p<0.001.

severity, timing of antivenom administration, antivenom protocol, thrombotic complications and follow-up were fully available (Table 1). Among the 116 envenomed patients, a total of 46 patients (39.6%) had have ADAMTS13 activity determination at admission before antivenom therapy. There were no differences in terms of clinical conditions, main snakebite characteristics and grade severity between patients who had or not ADAMTS13 activity determination (Table 1).

Main characteristics of the 46 patients with available ADAMTS13 activity determination are displayed Table 2.

The median age of these 46 patients was 49 years (IQR: 36–60). The median time from the snakebite to hospital admission was 2:17 hours (IQR: 1:51–3:30). Most participants were male (70%). The envenomed patients were agricultural workers in 30% of cases. Moderate envenoming was the most common presentation (78%). Two patients developed ischemic cerebral stroke. The median plasmatic ADAMTS13 activity was 94% (IQR: 78–122%), while an ADAMTS13 activity less than 75% was observed in ten (22%) patients. This group of 10 patients were older and displayed a lower platelet count (which remain normal), a more intense myonecrosis along with higher fibrinogen and C-reactive protein levels compared with patients with normal ADAMTS13 activity (Table 2).

Levels of main biological parameters of the 46 patients according to age of the patients are displayed S1 Table. No statistical differences were found between patients >65-year-old compared with patients <65-year-old).

Due to the small number of cerebral strokes in our series (n = 2 patients), the clinical relevancy of low ADAMTS13 levels as an independent factor of thrombotic complications was not investigated. While low ADAMTS13 levels has been consistently associated with end-organ failure, longer hospital stays and increased mortality in various pathologies, we further tested whether ADAMTS13 level as an independent predictor of length of hospitalized patients suffering *B. lanceolatus* envenoming. In line, a multivariate linear regression was performed to test the relationship between length of hospital stay and independent predictors (Table 3).

Variables were considered for multivariate analysis after taking into account their clinical pertinence and significant association (p<0.10) in univariate analysis. Variables entered into the initial multivariate linear model for length of hospital

**Table 1. Baseline characteristics of patients bitten by *B. lanceolatus*.**

| | All patients | Patients with ADAMTS13 | Patients without ADAMTS13 | P |
|---|---|---|---|---|
| **Number of cases** | **116** | **46** | **70** | |
| Age (years), IQR | 50 (36-60) | 49 (36-60) | 50 (39-60) | 0.919 |
| Gender (male), n (%) | 75 (64.7%) | 32 (69.6%) | 43 (61.4%) | 0.2434 |
| Cardiovascular risk factors, n (%) | 35 (30%) | 17 (37%) | 18 (26%) | 0.139 |
| **Snakebite presentation** | | | | |
| Agriculture, n (%) | 38 (33%) | 14 (30%) | 24 (34%) | 0.691 |
| Trekking, n (%) | 13 (11%) | 7 (10%) | 6 (13%) | 0.412 |
| Snake identified, n (%) | 28 (24%) | 13 (28%) | 15 (21%) | 0.266 |
| Upper limb site of bite, n (%) | 25 (22%) | 8 (17%) | 17 (24%) | 0.259 |
| Lower limb site of bite, n (%) | 91 (78%) | 53 (76%) | 38 (83%) | 0.259 |
| Delay for antivenom (hrs:min), IQR | 2:49 (2:00-5:05) | 2:17 (1:51-3:30) | 3:00 (2:10-5:40) | 0.225 |
| **Grade severity of envenoming** | | | | 0.094 |
| Grade 1, n (%) | 5 (4.3%) | 0 (0%) | 5 (7.1%) | – |
| Grade 2, n (%) | 92 (79.3%) | 36 (78.3%) | 56 (80.0%) | – |
| Grade 3, n (%) | 13 (11.2%) | 7 (15.2%) | 6 (8.6%) | – |
| Grade 4, n (%) | 6 (5.2%) | 3 (6.5%) | 3 (4.3%) | – |
| **Clinical features/ complications** | | | | |
| Systolic arterial pressure, IQR | 143(123-156) | 138 (129-151) | 146 (122-160) | 0.575 |
| Heart rate (bpm), IQR | 80 (69-90) | 81 (69-91) | 80 (69-90) | 0.989 |
| $SpO_2$ (%), IQR | 98 (97-100) | 98 (97-100) | 98 (98-100) | 0.515 |
| Temperature (°C), IQR | 36.7 (36.5-37.0) | 36.8 (36.6-37.1) | 36.4 (36.6-36.9) | 0.141 |
| Stroke, n (%) | 2 (2%) | 0 (0%) | 2 (2.9%) | 0.362 |
| **Biological analysis** | | | | |
| Hemoglobin (g/dL), IQR | 15.3 (14.3-16.5) | 15.1 (14.1-16.3) | 15.4 (14.6-16.8) | 0.767 |
| Leukocyte count ($10^3$/µL), IQR | 7.6 (5.8-9.8) | 7.7 (5.9-9.7) | 7.5 (5.8-9.9) | 0.287 |
| Platelet count (Giga/L), IQR | 240 (204-272) | 230 (199-275) | 243 (205-266) | 0.669 |
| Thrombocytopenia, n (%) | 5 (4.3%) | 2 (4.3%) | 3 (4.3%) | 0.990 |
| Partial thromboplastin time, PTT (%), IQR | 100 (93-107 | 98 (93-107) | 101 (93-109) | 0.759 |
| Prothrombin time, PT (%), IQR | 95 (87-106) | 94 (84-107) | 95 (89-106) | 0.872 |
| Fibrinogen (g/L), IQR | 3.2 (2.9-3.9) | 3.2 (2.9-3.8) | 3.3 (2.8-4.0) | 0.640 |
| C-reactive protein (mg/dL), IQR | 2.6 (0.7-7.9) | 2.5 (0.6-7.2) | 3.0 (0.7-9.8) | 0.129 |
| CPK (IU/L), IQR | 222 (122-377) | 229 (146-336) | 211 (116-465) | 0.647 |
| ADAMTS13 activity (%), IQR | | 92.5 (77.5-116.8) | | na |
| **Treatment** | | | | |
| Bothrofav administration, n (%) | 110 (95%) | 45 (98%) | 65 (93%) | 0.232 |
| Antibiotics, n (%) | 25 (22%) | 12 (26%) | 13 (19%) | 0.231 |
| **Outcome** | | | | |
| Ambulatory patients, n (%) | 57 (49%) | 22 (48%) | 35 (50%) | 0.484 |
| Length of hospital stay (day), mean±SD | 2.0±1.9 | 2.0±1.8 | 2.1±1.9 | 0.969 |
| Deceased patients, n (%) | 0 (0%) | 0 (0%) | 0 (0%) | na |

Results are reported as median and inter-quartile range (IQR) or as absolutes number of cases and percentages. Abbreviation: ADAMTS13, a disintegrin-like and metalloproteinase with a thrombospondin type 1 motif, member 13; CPK, creatine phosphokinase; MRI, magnetic resonance imaging; na, non-applicable. Thrombocytopenia was defined as a platelet count of less than 150 Giga/L.

**Table 2. Baseline characteristics of patients bitten by *B. lanceolatus* with ADAMTS13 activity determination.**

| | All patients | ADAMTS13>75% | ADAMTS13<75% | P |
|---|---|---|---|---|
| **Number of cases** | **46** | **36** | **10** | |
| Age (years), IQR | 49 (36-60) | 47 (34-59) | 58 (45-73) | 0.025 |
| Gender (male), n (%) | 32 (70%) | 23 (64%) | 9 (90%) | 0.143 |
| Cardiovascular risk factors, n (%) | 17 (37%) | 13 (36%) | 4 (40%) | 0.550 |
| **Snakebite presentation** | | | | |
| Agriculture, n (%) | 14 (30%) | 12 (33%) | 2 (20%) | 0.347 |
| Trekking, n (%) | 7 (10%) | 6 (17%) | 1 (10%) | 0.425 |
| Snake identified, n (%) | 13 (28%) | 9 (25%) | 4 (4(0%) | 0.289 |
| Upper limb site of bite, n (%) | 8 (17%) | 5 (14%) | 3 (30%) | 0.344 |
| Lower limb site of bite, n (%) | 38 (83%) | 31 (82%) | 7 (70%) | 0.344 |
| Delay for antivenom (hrs:min), IQR | 2:17 (1:51-3:30) | 2:17 (1:54-3:57) | 2:00 (1:10-3.14) | 0.301 |
| **Grade severity of envenoming** | | | | 0.208 |
| Grade 1, n (%) | 0 (0%) | 0 (0%) | 0 (0%) | |
| Grade 2, n (%) | 36 (78.3%) | 29 (80.6%) | 7 (70.0%) | |
| Grade 3, n (%) | 7 (15.2%) | 4 (11.1%) | 3 (30.0%) | |
| Grade 4, n (%) | 3 (6.5%) | 3 (8.3%) | 0 (0%) | |
| **Clinical features/ complications** | | | | |
| Systolic arterial pressure, IQR | 138 (129-151) | 138 (125-155) | 143 (129-158) | 0.581 |
| Heart rate (bpm), IQR | 81 (69-91) | 81 (69-94) | 70 (62-83) | 0.078 |
| SpO$_2$ (%), IQR | 98 (97-100) | 98 (97-100) | 97 (96-100) | 0.160 |
| Temperature (°C), IQR | 36.8 (36.6-37.1) | 36.8 (36.7-37.1) | 36.6 (36.6-37.1) | 0.393 |
| Stroke, n (%) | 0 (0%) | 0 (0%) | 0 (0%) | na |
| Peripheral thrombosis, n (%) | 1 (2.2%) | 1 (2.8%) | 0 (0%) | 0.783 |
| **Biological analysis** | | | | |
| Hemoglobin (g/dL), IQR | 15.1 (14.1-16.3) | 15.0 (14.0-16.3) | 15.4 (14.6-16.8) | 0.776 |
| Leukocyte count (10$^3$/µL), IQR | 7.7 (5.9-9.7) | 7,5 (5,9-9,6) | 7,8 (5,8-9,8) | 0.972 |
| Platelet count (Giga/L), IQR | 230 (199-275) | 235 (171-269) | 243 (205-266) | 0.047 |
| Partial thromboplastin time, PTT | 98 (93-107) | 97 (92-107) | 103 (92-109) | 0.649 |
| Prothrombin time, PT (%), IQR | 94 (84-107) | 95 (87-110) | 87 (74-106) | 0.352 |
| Fibrinogen (g/L), IQR | 3.2 (2.9-3.8) | 3.1 (2.9-3.5) | 4.2 (3.2-7.0) | 0.011 |
| C-reactive protein (mg/dL), IQR | 2.5 (0.6-7.2) | 2.4 (0.5-5.2) | 8.2 (0.6-8.0) | 0.020 |
| CPK (IU/L), IQR | 229 (146-336) | 223 (147-313) | 238 (104-621) | 0.026 |
| ADAMTS13 activity (%), IQR | 92.5 (77.5-116.8) | 105 (88-129) | 66 (62-74) | <0.001 |
| **Treatment** | | | | |
| Bothrofav administration, n (%) | 45 (98%) | 35 (97%) | 10 (10%) | 0.783 |
| Antibiotics, n (%) | 12 (26%) | 8 (22%) | 4 (40%) | 0.416 |
| **Outcome** | | | | |
| Ambulatory patients, n (%) | 22 (48%) | 18 (60%) | 4 (40%) | 0.725 |
| Length of hospital stay (day), mean±SD | 2.0±1.8 | 1.7±1.0 | 3.2±3.2 | 0.019 |
| Deceased patients, n (%) | 0 (0%) | 0 (0%) | 0 (0%) | na |

Results are reported as median and inter-quartile range (IQR) or as absolutes number of cases and percentages. Abbreviation: ADAMTS13, a disintegrin-like and metalloproteinase with a thrombospondin type 1 motif, member 13; CPK, creatine phosphokinase; MRI, magnetic resonance imaging; na, non-applicable.

**Table 3. Analysis of the relationship between length of hospital stay and independent predictors (patient and envenoming characteristics): univariate and multivariate linear regression.**

|  | Univariate analysis | | | Multivariate analysis | |
|---|---|---|---|---|---|
|  | B-coefficient | β coefficient | P value | β coefficient | P value |
| Age, years | 0.032 | 0.278 | 0.061 |  |  |
| Delay for antivenom | <0.001 | -0.068 | 0.698 |  |  |
| Grade severity of envenoming | 1.399 | 0.456 | 0.001 | 0.486 | 0.002 |
| Platelet count | 0.005 | -0.215 | 0.150 |  |  |
| Fibrinogen | 0.704 | 0.444 | 0.010 |  |  |
| CPK | 0.001 | 0.062 | 0.702 |  |  |
| C-reactive protein | 0.015 | 0.415 | 0.004 | 0.291 | 0.054 |
| ADAMTS13 activity | 0.016 | -0.259 | 0.082 | -0.285 | 0.057 |

Abbreviation: ADAMTS13, a disintegrin-like and metalloproteinase with a thrombospondin type 1 motif, member 13; CPK, creatine phosphokinase. Beta coefficients (β-coefficient), i.e., standardized regression coefficients, are displayed to compare the relative strengths of our predictors. B-coefficient (B-coefficient) indicates the average increase in length of hospital stay associated with a 1-unit increase in a predictor.

stay: age, grade of the envenoming, fibrinogen, CPK, C-reactive protein and ADAMTS13 activity. A backward stepwise regression approach was then applied to produce the final multivariate model, with a statistical significance level set at $p < 0.05$. Multivariate linear regression retained only grade severity of envenoming as independent predictor of increase length of hospital stay, while reduced ADAMTS13 activity and increased C-reactive protein levels bordered on statistical significance.

## Discussion

A syndrome consistent with thrombotic microangiopathy may develop within the few days after snakebite envenoming and combines features such as thrombocytopenia, hemolytic anemia with presence of schistocytes in blood smears, decreased haptoglobin level, increased lactate dehydrogenase level and acute kidney injury [15,16]. While thrombotic microangiopathy has been occasionally described in *Bothrops* sp. envenoming [17–19,29–32], only few case reports have previously evaluated the levels of ADAMTS13 [17,18,29–31], the main biological factor of endothelial dysfunction and thrombosis risk associated with this syndrome. For the first time, our study provided evidence suggesting that ADAMTS13 activity is reduced in experimental venom exposure and in human *B. lanceolatus* envenoming. While it was found that reduced ADAMTS13 activity <75% was not associated with more severe envenoming, our study results further suggest that, together with high grade severity score of envenoming and increased C-reactive protein, moderately reduced levels of ADAMTS13 activity are associated with increased length of hospital stays.

ADAMTS13 is primarily synthesized in the liver, and its main function is to cleave von Willebrand factor (VWF) anchored on the endothelial surface, in circulation, and at the sites of vascular injury [20–23]. The formation and elongation of VWF strings on endothelial surfaces are tightly regulated by ADAMTS13 through proteolytic cleavage of VWF strings. In the absence or reduction of ADAMTS13 activity, ultra-large von Willebrand factor strings remain anchored at the endothelium and can recruit flowing platelets and causing uncontrolled thrombosis in terminal arterioles and capillaries [20–23]. Along with endothelial inflammation and recruitment of immune competent cells, such deregulation of the ADAMTS13-vWF axis can play a critical role in the pathophysiology of thrombotic complications observed in human *B. lanceolatus* envenoming.

Our experimental results provide the new information that incubation of human plasma with known elevated ADAMTS13 activities with *B. lanceolatus* venom dose dependently reduced ADAMTS13 activity. In line, findings in

*Bothrops* sp. envenoming have previously reported mild reduction of ADAMTS13 activity in rodents [26,27]. Such observations have been attributed to the proteolytic cleavage of ADAMTS13 by snake venom metalloproteinases (SVMP), because Na$_2$-EDTA (a SVMP inhibitor) may partially block these alterations [26]. It has been also shown that enzymatic activities of thrombin, FXa, and plasmin, which typically generated in the systemic circulation during *Bothrops* sp. envenoming, can induced a proteolytic inactivation of ADAMTS13 [33]. Incubation of ADAMTS13 with varying concentrations of human thrombin, followed by Western blotting with the anti-ADAMTS13 protease domain antibody, revealed that ADAMTS13 was cleaved by thrombin in a time- and concentration-dependent fashion. Similarly, experiments performed using FXa and plasmin demonstrated that the factors caused ADAMTS13 fragmentation similarly to that caused by thrombin [33,34]. In addition to proteolytical inactivation [33,34], conformational changes of ADAMTS13, i.e., closed or open, have been consistently shown to alter its activity [35,36]. For example, disruption of the closed conformation of ADAMTS13 contributes to increased ADAMTS13 activity toward VWF [35].

Our clinical study suggests that some patients suffering from *B. lanceolatus* envenoming displayed moderately low ADAMTS13 activity. According to our results, it was considered that reduced ADAMTS13 activity <75% was not associated with more severe envenoming. However, compared with patients with normal ADAMTS13 activity, those with moderately low ADAMTS13 activity have no major changes in coagulation tests but displayed biological evidence of systemic inflammatory, such as higher fibrinogen and C-reactive protein levels. An increased risk of thrombosis is expected in patients with moderately low ADAMTS13 activity due to the imbalance of the ADAMTS13/VWF axis (decreased levels of ADAMTS13 and increased levels of VWF multimers), which exacerbates the thromboinflammatory response and in turn causes hypercoagulation, inhibition of fibrinolysis, neutrophil activation, and interaction with neutrophil extracellular traps [37–40]. Hence, it is proposed that reduced ADAMTS13 activity observed in some *B. lanceolatus* envenomed patients can be associated with an increased risk of thrombotic events.

New information provided by our study emphasizes prognosis factors that may independently and collectively pose a risk to *B. lanceolatus* envenoming patients [41,42]. Previously published risk factors of increased hospital stays and poor outcome in adults include delay from the bite to adequate antivenom therapy, acute renal failure and abnormal blood results such as thrombocytopenia, coagulation test derangements (prothrombin time > 13.2 s, partial thromboplastin time > 37.2 s, low fibrinogen levels), elevated creatinine kinase (CK), along with organ failure (respiratory failure, shock, neurotoxicity, ischemic stroke) associated with specific snake species [43–45]. Several scoring systems have been developed to risk-stratify patients with snakebite envenoming. The Snakebite Severity Score is an objective symptom severity scoring tool developed and validated for the evaluation of initial severity and crotaline envenomation progression and to guide antivenom therapy [44]. This score provides a comprehensive framework based on clinical assessment and coagulation tests to facilitate risk stratification of envenoming [43–45]. In our experience in Martinique, we used a simple scale to grade patients bitten by *B. lanceolatus*, which focuses on signs such as swelling, pain, and systemic manifestations [41,42]. Minor presentations on this scale are grade 1, while severe swelling with or without systemic features is assigned to grade from 2 to 4 guide decision and dose for antivenom [6,39,40].

In the present study, the number of thrombotic events, which may be mechanistically related to low ADAMTS13 levels, was quite low. Such a small sample size (n = 2) precluded any consistent analysis aimed to identify low ADAMTS13 levels as an independent predictor of thrombotic events in patients suffering *B. lanceolatus* envenoming. Because low ADAMTS13 levels has been consistently associated with end-organ failure, longer hospital stays and increased mortality in various pathologies [24,25,46–48], we investigated whether low ADAMTS13 level was an independent predictor increased length of hospital stay. The present study confirms that the grading severity system used in Martinique was an independent predictor of length of hospital stay, while reduced ADAMTS13 activity and increased C-reactive protein levels bordered on statistical significance. Of note, advanced age is one of the main factors affecting the levels of many biological parameters. In our series, patients with reduced ADAMTS13 activity were older than those with normal activity. Hence, age (≥65-year-old) could have contributed to aberrant plasma indicators of endothelial coagulopathy such as ADAMTS13

[49]. Hence, while normal levels of biological markers for a healthy elderly people should be considered, no difference was found between patients <65-year-old and patients <65-year-old in our study, suggesting that reduced levels of biomarkers including ADAMTS13 was because of the snake venom, but not related to advanced age.

### Limitations of the study

Several study limitations are to be noted in our exploratory research. Few reports have previously provided range of plasmatic venom concentration in *B. lanceolatus* envenoming. In our *in vitro* experimental studies, we have used venom concentration in the range of 10–1000 ng/mL, which is in line with those previous observations [39]. Of note, our *in vitro* experiments clearly suggest that reduced ADAMTS13 activity was a unique feature of *B. lanceolatus* venom.

One significant concern pertains to the data collection process in this retrospective series of patients hospitalized at the University Hospital of Martinique. All data obtained in the present study was subject to review by the healthcare professionals of the snakebite group at the University Hospital of Martinique responsible for recording clinical information. Another main limitation is the small sample size of our study. Hence, the clinical consequences of reduced ADAMTS13 activity in *B. lanceolatus* envenoming was not studied in term of thrombosis.

In this series of patients, specific outcome related to ischemic events, i.e., stroke, was not considered since only two cases of stroke were recorded. This issue represents a major concern because one of the main clinical complications of *B. lanceolatus* envenoming are thrombotic events in the course of the hospitalization. While reduced ADAMTS13 activity has been consistently linked to increased thrombosis risk, our study did not demonstrate a higher incidence of thrombotic events in patients with reduced ADAMTS13.

Regarding the mechanisms of ADAMTS13 reduction, biological testing to document microangiopathic hemolytic anemia was not performed in our study. In our experience in Martinique, this specific complication is not observed in the case of *B. lanceolatus* envenoming. Typically, envenomed patients presented with normal hemoglobin concentration, normal renal function and absence of schistocytes on the automatized microscopic examination of peripheral blood smear. When considering length of hospital stay as an outcome, our results identified envenoming severity as an independent predictor, but reduced ADAMTS13 activity only bordered on statistical significance in the multivariate linear regression, possibly due to the small sample size of our study. In line with the reduced ADAMTS13 activity, determination of the level of plasmatic ultra-large von Willebrand multimer (ULVWF) was not available in this retrospective study. Indeed, increased ULVWF in the plasma as a consequence of would ascertain the prothrombotic role of reduced ADAMTS13 activity in *B. lanceolatus* envenoming.

Eventually, whether reduced ADAMTS13 activity is a specific feature of *B. lanceolatus* venom should be documented in humans by comparison with *Bothrops* sp. envenoming typically associated with consumption coagulopathy.

### Conclusion

For the first time, our study provided evidence suggesting that ADAMTS13 activity is reduced in experimental venom exposure and in human *B. lanceolatus* envenoming. Such moderately low ADAMTS13 activity may be implicated in the increased risk of thrombosis *B. lanceolatus* envenomed patients due to the imbalance of the ADAMTS13/VWF axis, which exacerbates biological features of thromboinflammation. Our study results further suggest that, together with high grade severity score of envenoming and increased C-reactive protein, moderately reduced levels of ADAMTS13 activity are associated with increased length of hospital stays.

### Supporting information

**S1 Table. Levels of main biological parameters of the 46 patients according to age of the patients.**
(DOCX)

**S1 Data. Raw data (Excel file).**
(XLS)

## Author contributions

**Conceptualization:** REMI NEVIERE, Dabor RESIERE.

**Data curation:** Jonathan FLORENTIN, Caroline RAPON, Olivier PIERRE-LOUIS, Fatima RADOUANI, Prisca JALTA, Pascal FUSEAU, Hatem KALLEL, Dabor RESIERE.

**Formal analysis:** REMI NEVIERE.

**Writing – original draft:** REMI NEVIERE, Dabor RESIERE.

## Acknowledgments

The authors declare that generative AI and/or AI-assisted technologies have not been used in the writing process.

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
