## [Decision Letter · Decision Letter 0]

15 Oct 2025

ADAMTS-13 activity is reduced by Bothrops lanceolatus snake venom: in vitro experiments and clinical correlation

Dear Dr. NEVIERE,

Thank you for submitting your manuscript to PLOS Neglected Tropical Diseases. After careful consideration, we feel that it has merit but does not fully meet PLOS Neglected Tropical Diseases's publication criteria as it currently stands. Therefore, we invite you to submit a revised version of the manuscript that addresses the points raised during the review process.

Please submit your revised manuscript within 60 days Dec 14 2025 11:59PM. If you will need more time than this to complete your revisions, please reply to this message or contact the journal office at plosntds@plos.org. Please include the following items when submitting your revised manuscript:

We look forward to receiving your revised manuscript.

Kind regards,

Wuelton Monteiro, Ph.D.

Section Editor

Wuelton Monteiro

Section Editor

Shaden Kamhawi

co-Editor-in-Chief

Paul Brindley

co-Editor-in-Chief

**Journal Requirements:**

At this stage, the following Authors/Authors require contributions: REMI NEVIERE. Please ensure that the full contributions of each author are acknowledged in the "Add/Edit/Remove Authors" section of our submission form.

- ® on pages: 4, 5, 6, 7, 10 and 11.

5) Please ensure that the funders and grant numbers match between the Financial Disclosure field and the Funding Information tab in your submission form. Note that the funders must be provided in the same order in both places as well.

**Reviewers' Comments:**

Reviewer's Responses to Questions

**Key Review Criteria Required for Acceptance?**

**Methods:**

-Are the objectives of the study clearly articulated with a clear testable hypothesis stated?

-Is the study design appropriate to address the stated objectives?

-Is the population clearly described and appropriate for the hypothesis being tested?

-Is the sample size sufficient to ensure adequate power to address the hypothesis being tested?

-Were correct statistical analysis used to support conclusions?

-Are there concerns about ethical or regulatory requirements being met?

Reviewer #1: YES

Reviewer #2: (No Response)

**Results:**

-Does the analysis presented match the analysis plan?

-Are the results clearly and completely presented?

-Are the figures (Tables, Images) of sufficient quality for clarity?

Reviewer #1: YES

Reviewer #2: (No Response)

**Conclusions:**

-Are the conclusions supported by the data presented?

-Are the limitations of analysis clearly described?

-Do the authors discuss how these data can be helpful to advance our understanding of the topic under study?

-Is public health relevance addressed?

Reviewer #1: YES

Reviewer #2: (No Response)

**Editorial and Data Presentation Modifications?**

Reviewer #1: MAJOR REVISION

Reviewer #2: (No Response)

**Summary and General Comments:**

Reviewer #1: This study investigates the impact of Bothrops lanceolatus venom on ADAMTS-13 activity. The research includes both in vitro experiments and a clinical study of patients bitten by this snake. The findings suggest that B. lanceolatus venom reduces ADAMTS-13 activity in a dose-dependent manner, potentially increasing the risk of thrombosis in envenomed patients.

While the authors addressed a very interesting topic, I have some concerns that are detailed below:

Title Accuracy: The title “ADAMTS-13 activity is reduced by Bothrops lanceolatus snake venom: in vitro experiments and clinical correlation” does not reflect the use of B. atrox venom in the in vitro experiments. Please revise the title to accurately represent the study's scope or remove the in vitro experiments involving B. atrox venom.

Objective: In the main objectives, the study was designed to test whether plasmatic ADAMTS-13 activity would be reduced by in vitro exposure to B. lanceolatus venom (in contrast with B. atrox venom) and in humans bitten by B. lanceolatus snake. This objective refers to both in vitro and human experiments. However, the method section lacks detailed information about the in vitro experimental design. Please provide a comprehensive description of the in vitro methodology.

Also, please clearly detail the primary objective and any secondary objectives of this study.

P6 – L163-164: The sentence “Patients with the diagnosis of B. lanceolatus envenoming presenting at the University Hospital of Martinique from January 2019 to Janvier 2023 were retrospectively.” Is incomplete. Please correct

In the method section, authors said that they used crude venoms obtained from adult wild-caught B. lanceolatus (n=12) and B. atrox specimens 139 (n=5), captured in Martinique and French Guyana, respectively. Please provide further detail on the composition of these venoms.

In the result section, authors presented the “In vitro effects of B. lanceolatus venom on plasmatic ADAMTS13 activity”. While, the invitro study method was not clearly defined and the reported results refers to both B. lanceolatus and B. atrox.

Authors also suggest that reduced ADAMTS-13 activity was associated with more severe envenoming. However, this assertion is inconsistent with the data, as 70% of envenomed patients with ADAMTS-13 < 75% belong to Grade 2, and no patients in Grade 4 had ADAMTS-13 activity < 75%. Please reconcile this discrepancy.

Authors performed a multivariate linear regression to test independent predictors of length of hospital stay. Please explain the rationale for conducting this comparison, as it does not appear to align with the stated objectives of the study. It feels out of the scope and doesn't significantly enhance the results.

Same remark on the discussion section where authors discuss factors associated to unfavorable outcome (P14 – L 315 to P15 – L 334). This is not in the objective of the study and doesn't contribute substantially to the reader's understanding of the primary findings.

In the result section, table 1 and 2, authors give patients characteristics but omit details on biological renal function, haptoglobin, schistocytes, LDH, and factor VIII level. These parameters are essential to interpret the data presented in this study and explain the precise mechanism of the envenoming.

Figure 1: ADAMTS-13 activity in serum exposed to 1000 ng/ml B. atrox venom was lower than in serum exposed to 10 and 100 ng/ml. Please check this result for accuracy and give explanation. Furthermore, given that the primary focus of this study is the effect of B. lanceolatus venom on ADAMTS-13 activity, the concurrent investigation of B. atrox venom is potentially confusing. These two venoms exhibit different effects on coagulation and induce distinct clinical manifestations, which could dilute the clarity of the core findings.

Table 1: the variable “ADAMTS13 activity <75%” is redundant since Q1 for ADAMTS13 activity was 77.5%

While reduced ADAMTS-13 activity is linked to increased thrombosis risk, the study doesn't directly demonstrate a higher incidence of thrombotic events in patients with reduced ADAMTS-13. Please explain this limitation in detail.

Could authors elaborate on the potential mechanisms by which B. lanceolatus venom reduces ADAMTS-13 activity?

P6 - L164 and Page 8 - L 217: please change “janvier” to “January”

In the whole manuscript: Authors used Adamst13, ADAMTS13, and ADAMTS-13. Please use ADAMTS-13 for uniformity.

Please discuss the limitations of the study more explicitly.

Reviewer #2: (No Response)

PLOS authors have the option to publish the peer review history of their article (what does this mean? ). If published, this will include your full peer review and any attached files.

**Do you want your identity to be public for this peer review?** For information about this choice, including consent withdrawal, please see our Privacy Policy .

Reviewer #1: No

Reviewer #2: No

**Figure resubmission:**
---

## [Editor Report · Decision Letter 1]

22 Oct 2025

Dear Pr NEVIERE,

We are pleased to inform you that your manuscript 'ADAMTS13 in Bothrops lanceolatus snakebite envenoming: crude venom-induced reduction of in vitro enzymatic activity and clinical correlation in snakebite patients' has been provisionally accepted for publication in PLOS Neglected Tropical Diseases.

Best regards,

Wuelton Monteiro, Ph.D.

Section Editor

Wuelton Monteiro

Section Editor

Shaden Kamhawi

co-Editor-in-Chief

Paul Brindley

co-Editor-in-Chief

---

## [Editor Report · Acceptance letter]

Dear Pr NEVIERE,

We are delighted to inform you that your manuscript, "ADAMTS13 in Bothrops lanceolatus snakebite envenoming: crude venom-induced reduction of in vitro enzymatic activity and clinical correlation in snakebite patients," has been formally accepted for publication in PLOS Neglected Tropical Diseases.

Best regards,

Shaden Kamhawi

co-Editor-in-Chief

Paul Brindley

co-Editor-in-Chief
